# Perspectives in the Diagnosis, Clinical Impact, and Management of the Vulnerable Plaque

**DOI:** 10.3390/jcm14051539

**Published:** 2025-02-25

**Authors:** Alberto Alperi, Paula Antuna, Marcel Almendárez, Rut Álvarez, Raquel del Valle, Isaac Pascual, Daniel Hernández-Vaquero, Pablo Avanzas

**Affiliations:** 1Hospital Universitario Central de Asturias, Oviedo, 33011 Asturias, Spain; alberto.alperi.garcia@hotmail.com (A.A.); paulantu@hotmail.com (P.A.); marcel.almendarez@gmail.com (M.A.); rutalvarez3@gmail.com (R.Á.); ipascua@live.com (I.P.); dhvaquero@gmail.com (D.H.-V.); 2Instituto de Investigación Sanitaria del Principado de Asturias (ISPA), 33011 Asturias, Spain; 3Faculty of Medicine, University of Oviedo, 33011 Asturias, Spain; 4Centro de Investigación en Red de Enfermedades Cardiovasculares (CIBERCV), 28046 Madrid, Spain

**Keywords:** vulnerable plaque, percutaneous coronary intervention, coronary artery disease

## Abstract

Coronary artery disease is a highly prevalent disease that constitutes the leading cause of mortality worldwide. Acute coronary syndromes are the most devastating form of presentation of coronary disease, involving the acute formation of a thrombus within the coronary vessel lumen, further leading to flow limitation and diminished myocardial perfusion. Vulnerable plaques, which are characterized by thin-cap fibroatheroma, a large lipid pool, and macrophage infiltration and spotty calcification of the cap, pose a higher risk of coronary events despite not being flow-limiting. Iterations in intravascular imaging and coronary computed tomography have largely increased the ability to detect and define vulnerable plaques, and its clinical impact in early- and mid-term outcomes has been confirmed in several studies. In this review, we aimed to revise the current concept of vulnerable coronary plaque and its repercussion, to summarize the main pharmacological approaches for its management, and to provide an updated overview of the available evidence on preventive percutaneous interventional strategies in this clinical setting.

## 1. Introduction

Atherosclerosis is a progressive pathological condition affecting over 500 million people globally. Its prevalence is expected to rise in the coming decades due to aging populations and lifestyle changes. Nowadays, atherosclerotic processes involving the coronary arteries, known as coronary artery disease (CAD), is yet considered as the leading cause of death overall [1].

CAD develops gradually through several steps, primarily driven by chronic inflammation of the coronary arterial walls. First, a continuous aggregation of low-density lipoprotein (LDL) particles leading to intima layer thickening occurs. This is followed by the infiltration of macrophage, further leading to macrophagic necrosis and apoptosis resulting into a lipid-rich necrotic core devoid of cellular content. Over time, this conglomerate is encapsulated by fibrous tissue, and calcification may occur at any evolving point [2,3].

Acute coronary syndromes (ACS) are the most dreadful manifestation of CAD, frequently involving the acute formation of a thrombus within the coronary vessel lumen, further leading to flow limitation and diminished myocardial perfusion. Currently, there is knowledge of three types of mechanistic processes resulting in acute intracoronary thrombus formation: plaque rupture, plaque erosion, and calcified nodule. Plaque rupture consists in a gap within the fibroatheroma plaque enabling direct contact between thrombogenic components of the plaque necrotic core and blood flow, what leads to thrombus formation. Plaque erosion is defined by the absence of endothelial coverage at a certain point, hence facilitating the contact between smooth muscle cells and coronary blood flow. Finally, calcified eruptive nodules are those with irregular surface lacking an overlying fibrous cap [4]. The first mechanism accounts for almost 80% of ACS cases in clinical practice [5].

Ancillar approaches for treating CAD aimed to restore blood flow in obstructive and flow-limiting lesions by means of percutaneous coronary interventions (PCI) and bypass graft surgery. High-risk atherosclerotic plaques prone to thrombus formation and ACS events—referred to as “vulnerable plaques”—often differ from their stable counterparts. These plaques are typically eccentric, angiographically non-severe, non-flow-limiting, and exhibit distinct histological characteristics compared to low-risk stable atherosclerotic plaques. Their non-stenotic and apparently safe aspect often prevents early detection. However, continuous technological iterations have provided more accurate diagnostic tools that favored the understanding of vulnerable plaque composition and clinical impact.

In this review, we aimed to revise the current concept of vulnerable coronary plaque and its repercussions, to summarize the main pharmacological approaches for its management, and to provide an updated overview of the available evidence on percutaneous interventional strategies in this clinical setting.

## 2. Vulnerable Plaque Definition

The term “vulnerable plaque” has been introduced decades ago in an attempt to define the characteristics of coronary atherosclerotic plaques prone to rupture. Over the years the concept has evolved, as it has been proposed also as a lesion conferring a high predisposition for thrombosis [6] and, lately, as a plaque which provides a higher risk for future major adverse cardiac events (MACE) [7]. Nevertheless, these concepts are closely interconnected, and, since plaque rupture remains as the most common mechanism for acute thrombus formation in atherosclerotic plaques, a vulnerable plaque can still be defined as one that is prone to rupture, thereby triggering thrombotic events and ACSs. Epidemiologically, plaque rupture accounts for 73% of coronary thrombi leading to sudden cardiac death [5]. Angiographically it cannot be differentiated from stable chronic plaques, but histological distinguished features associated with vulnerable plaques are well-known and presented below.

### Histological Definition

Histopathologically, a vulnerable plaque mainly consists of a large necrotic core with minimal or null cellularity, surrounded by a thin fibrous cap, which altogether is better known as thin-cap fibroatheroma (TCFA). This concept stems from autopsy histological studies, which demonstrated that there was an inverse relationship between cap thickness and the risk for plaque rupture, with a thickness of 65 πm usually used as a threshold for defining high-risk plaques [8]. Although fibrous cap thickness is the main driver for differentiating between stable and potentially vulnerable plaques, other characteristics have been intimately associated with plaque vulnerability: the dimension of the necrotic core (with larger necrotic cores posing a higher risk of instability); macrophage foam cell infiltration of the cap [2,9]; and spotty microcalcification of the thin fibrous cap [10], in opposition to stable larger calcific areas. Additionally, vulnerable plaques are more commonly associated with a positive remodeling (expansion) of the vessel wall, in contrast to the negative remodeling (reduction) frequently found in stable plaques [11]. As a consequence, vulnerable plaque lesions are more frequently unnoticed in coronary angiographies, as the lumen profile of the vessel remains almost unchanged.

## 3. Vulnerable Plaque Diagnosis

### 3.1. Invasive Imaging

Although the concept of plaque vulnerability emerged from autopsy studies, cardiac imaging (both intravascular and non-invasive) plays a key role for its early diagnosis in in vivo analysis.

Coronary angiography: coronary angiography is a mere lumenogram of the vessel lumen. While it is highly effective in detecting stenotic segments in orthogonal projections and remains one of the gold standards for diagnosing CAD, its sensitivity and specificity for identifying more subtle features—such as small intravascular thrombi, plaque rupture or erosion, and vulnerable plaques—are limited. This represents one of its major drawbacks [12].

Over the last decades, technological advancements have enabled to assess vessel wall components from the inner lumen. Each technique provides distinct advantages and limitations compared to its counterparts. The main characteristics for the intravascular imaging techniques with available clinical evidence in the evaluation of vulnerable plaques are displayed in Table 1.

Intravascular ultrasound (IVUS): this technique is based on an intracoronary catheter equipped with a piezoelectric crystal for ultrasound emission and post-processing. The assessment of atheromatous plaque based on grayscale IVUS is limited due to its low spatial resolution (axial resolution of circa 200 µm) and grayscale presentation. Therefore, it cannot provide detailed evaluation of lipid-rich plaques with TCFA [13].

In order to provide a more comprehensive assessment of plaque composition, IVUS post-processing methods have been developed over time. Virtual histology (VH)-IVUS (Philips Volcano Corp., California, USA) uses frequency in combination with echography intensity to provide a histological compound estimation that classifies atherosclerotic plaques in four types: fibrous, necrotic, calcific, and fibrofatty, each of them displayed in a default different color. Although VH-IVUS has been validated for identifying the components of a plaque, including areas of lipid-rich necrotic core [14], it has major limitations when aiming to evaluate thin fibrous cap thickness and composition due to spatial resolution issues. An alternative definition of thin fibrous cap has been proposed for VH-IVUS examinations, relying on the presence of a > 10% necrotic core, > 40% plaque burden, and the absence of overlying fibrous tissue. Nonetheless, some TCFA are yet misclassified with such approach [15].

Recently, high-definition IVUS based on 60 MHz technology (as opposed to 20–45 MGz of standard prior grayscale IVUS catheters) allows for a greater spatial resolution (around 40–60 µm) of the vessel wall components with a slightly decrease in tissue penetration. Future studies aiming to compare this promising technology with histological findings and to assess its utility in identifying components of plaque vulnerability are eagerly awaited. Near-infrared fluorescence IVUS, intravascular photoacoustic IVUS, and time-resolved fluorescence spectroscopy IVUS are novel techniques under investigation that might expand over the future.

Optical coherence tomography (OCT): OCT utilizes near-infrared light projected towards the vessel wall by means of a rotating optical fiber. The amplitude and delay of the backscattered light enables the formation of high-resolution cross-sectional and volumetric images of the vessel structure. The shorter wavelength of the infrared light in OCT (1.3 μm) confers greater axial resolution (10–20 μm versus 50–150 μm) but lower penetration depth (1–2 mm versus 5–6 mm) compared to the IVUS technology [16]. A good association between histological autopsy results and OCT findings for the characterization of several types of atherosclerotic plaques has been reported [13].

Due to its high spatial resolution, OCT is the only approach enabling a comprehensive assessment of the actual thickness of the fibrous cap. Additionally, lipid-rich plaques are characterized by high-attenuating, signal-poor regions covered with a signal-rich band that corresponds to the fibrous cap. Although debatable, there are established thresholds for defining an atherosclerotic plaque as vulnerable based on OCT criteria: thin fibrous cap thickness < 75 πm; lipid pool arc extension > 180° in a cross-sectional view, being the lipid pool defined as a signal-poor region diffusely bordered by overlying signal-rich fibrous cap; and the presence of macrophage clusters defined at visual estimation as signal-rich, distinct, or confluent punctate regions [17]. Additionally, vasa-vasorum and micro vessels within coronary atherosclerotic areas, which can be visualized by OCT, have also been linked to plaque vulnerability [18].

Near-infrared spectroscopy (NIRS): Spectroscopic analysis of the backscattered light emitted by an NIR probe informs on the extent of cholesterol within the arterial vessel wall. There are distinctive features associated with cholesterol in the spectrum of the wavelength used in NIRS, allowing for distinction of lipid core plaques. The obtained data is displayed as a chemogram of the probability of the presence of a lipid core plaque per millimeter of vessel length (with yellow images representing a higher probability). The lipid core burden index (amount of lipid in a scanned artery) is also provided, frequently indexed to a 4 mm length [19].

Combination of techniques: Since NIRS lacks the ability to inform on lumen area and plaque depth, this technique is currently frequently combined with an IVUS catheter with which both techniques can be performed simultaneously. Hybrid OCT/IVUS are already available in some countries, and its clinical use is expected to spread significantly over the upcoming years [20]. Besides, tri-combined catheters are under development and could provide an all-in-one tool for plaque assessment [21].

### 3.2. Non-Invasive Imaging

Coronary computed tomography (CCT): CCT has a high spatial resolution enabling an optimal visualization of the coronary tree. Although its main utility relies on its negative predictive value aiming to exclude subjacent CAD, some features associated with atherosclerotic plaque vulnerability might be distinguished by CCT scan. Low attenuation plaques (i.e., those exhibiting low Hounsfield Units) have been correlated with lipid-rich plaques when compared with VH-IVUS [22] and OCT [23], and can be easily distinguished from primarily calcific atherosclerosis. Nonetheless, lipidic and fibrotic plaques are hardly distinguishable by CCT in terms of tissular density, what partially limits its accuracy for plaque composition discrimination [24]. As for intravascular imaging, some features like vessel positive remodeling [25] and spotty calcification [26] can be evaluated by CCT and are associated with plaque vulnerability. Finally, the napkin-ring sign is another CCT-derived feature that has been associated with plaque vulnerability. It consists in a low attenuation area (probably a necrotic core) surrounded by a higher-attenuation ring, and it has been correlated with cap thickness [27,28].

Positron emission tomography (PET): PET in combination with CCT has been increasingly studied for the stratification of patients with known or at risk for CAD. 18F-fluorodeoxyglucose is a radiolabeled glucose analogue whose uptake is linked to high metabolic activity. In the context of CAD, it has been associated with culprit lesions in the setting of ACSs [29]. 18F-sodium fluoride is considered a better tracer for coronary disease, and it has also been evaluated as a marker of plaque instability and progression. Though it has been correlated with microcalcification and fibroateroma lesions in autopsy studies [30], the clinical impact of high uptake of this tracer remains controversial [31].

Cardiac magnetic resonance imaging (CMRI): Cardiac magnetic resonance imaging (CMRI) is a radiation-free imaging technique that allows vessel lumen visualization in the presence of calcium and enables accurate characterization of tissues. However, due to its limited spatial resolution and its modest diagnostic accuracy, it is not a widespread technique for assessing CAD.

### 3.3. Novel Markers

Novel biochemical markers might contribute to the detection of vulnerable plaques aside from invasive or non-invasive imaging. The CD93 molecule has been identified as a key mediator of endothelial inflammation and intraplaque neovascularization, factors that are well-known to contribute to plaque instability. Besides, macrophages with high CD93 expression have shown a greater capacity to infiltrate atheromatous plaque, which may contribute as well to its progression [32]. Recent research in murine models have suggested that CD93 signal could be used to improve the detection of unstable plaques by molecular imaging techniques (such as PET) [33].

### 3.4. Clinical Impact of Vulnerable Plaques

Autopsy studies have demonstrated the association between TCFA and myocardial infarction or sudden cardiac death, with an inverse relationship between cap thickness and the risk of atherosclerotic plaque rupture [2,34].

Several intravascular studies based on VH-IVUS have highlighted the clinical impact of plaque vulnerability in lesions that would otherwise be considered non-significant. In the PROSPECT study, 313 patients out of 697 participants had features of TCFA, and those patients exhibiting TCFA and a plaque burden > 70% in non-culprit lesions at baseline had a notable increased risk (HR 10.83, 95%CI 5.55–21.1) of major cardiac events (i.e., composite of cardiac death, myocardial infarction, and admission due to unstable angina) after a median follow-up period of 3.4 years [35]. In the same line, the VIVA trial demonstrated that VH-IVUS-derived TCFA and its extent were the only baseline factors associated with adverse coronary events related with non-culprit lesions during follow-up [36]. Finally, the ATHEROREMO-IVUS study showed that when IVUS-based TCFA is present in a non-culprit lesion there is a higher expected incidence of cardiac death/ACS at 1 year follow-up (adjusted HR 2.54, 95%CI 1.17–5.51) [37].

OCT-derived studies do also stress the prognostic significance of vulnerable plaques in CAD. The CLIMA study included 1003 patients undergoing clinically indicated coronary angiography with non-obstructive lesions in the proximal left anterior descending artery. Those lesions exhibiting high-risk features identified on OCT (such as a minimum lumen area [MLA] < 3.5 mm^2^, fibrous cap thickness < 75 µm, lipid arc > 180°, and macrophages) demonstrated a 7.5-fold higher risk of cardiac death or target segment-related myocardial infarction at 1 year compared to patients with non-vulnerable features [38]. The COMBINE OCT-FFR trial showed that among patients with ≥1 fractional flow reserve (FFR)-negative lesions, TCFA-positive patients (OCT-defined fibrous cap thickness of ≤65 μm with a lipid arc of >90°) represented 25% of this population and were associated with a fivefold higher rate of MACE despite the absence of ischemia [39]. Two additional large observational studies [40,41] demonstrated that the presence of vulnerable plaque characteristics detected by OCT was associated with an increased risk of MACE. Figure 1 displayed the rates of major adverse vascular events for patients with and without vulnerable plaques among the main IVUS and OCT studies.

## 4. Medical Therapies for the Treatment of Vulnerable Plaques

Atherosclerosis is a diffuse and systemic process. Although only a small proportion of all atherosclerotic plaques exhibit vulnerable features, systemic therapies targeting cholesterol, low-density lipoprotein (LDL) levels, and chronic inflammation have been associated with beneficial effects on vulnerable plaques and coronary atherosclerosis.

### 4.1. LDL-Lowering Drugs

Statins: The benefit of this pharmacological group in patients with established CAD has been largely defined, due to its lipid-lowering and pleiotropic effects [42]. Several NIRS- and IVUS-derived studies have evaluated the effect of statins on plaque progression in non-culprit intermediate lesions after an ACS [43,44] and in patients presenting with stable CAD [45,46]. Overall, a consistent reduction of plaque burden has been observed across all studies and, importantly, the reduction of the progression of coronary atherosclerosis seems to be dependent on the intensity of the statin therapy, with higher intensities implying a greater plaque reduction and stabilization [46,47]. More specifically, pooled data of studies including IVUS-derived high-risk atherosclerotic plaques (i.e., those showing large atheroma volume, positive remodeling, and spotty calcification) has demonstrated that high-risk plaques exhibited the largest stabilization features in terms of total atheroma burden regression and diffuse non-spotty calcification disappearance [48] after initiation of statins.

OCT-derived studies are in line with prior findings, demonstrating an accurately measured increment of the fibrous cap and a reduction of the lipid-arch, both of which were directly proportional to the dose of statins and to the promptness of the initiation of such therapy [49,50].

Ezetimibe: this molecule has demonstrated its added value on top of statins for secondary cardiovascular prevention [42]. Additionally, it has been shown in IVUS dedicated studies that negative vessel remodeling and coronary plaque regression was greater with the combination of ezetimibe and statins than with statins in monotherapy [51]. This concept has also been suggested after OCT evaluation, mainly based on an increased fibrous cap thickness over time [52].

PCSK9 inhibitors: Proprotein convertase subtilisin–kexin type 9 inhibitors (PCSK9i) facilitates the degradation of LDL receptors within hepatocytes. This relatively novel therapy has demonstrated a huge reduction in LDL levels compared to prior available treatments, as well as a significant reduction in cardiovascular outcomes [53]. In regard to plaque vulnerability features, these molecules have demonstrated a reduction in NIRS- and IVUS-based atheroma burden at follow-up [54,55,56]. OCT findings in patients receiving Evolocumab for non-culprit moderate lesions (20–50% by angiography) demonstrated that minimum fibrous cap thickness increased, and both lipid arch and macrophage infiltration decreased after a follow-up period of 4 years [57]. Besides, in a CCT-based study of Evolocumab for non-significant coronary lesions, the utilization of this PCSK9i demonstrated a reduction in vulnerability characteristics primarily based in an increased plaque density and a reduction in remodeling index [58]. The main studies evaluating the impact of novel LDL-lowering therapies in coronary plaque features are summarized in Table 2.

Omega-3 fatty acids: eicosapentaenoic acid (EPA) is a novel therapy based on the anti-inflammatory and anti-oxidative properties of omega-3. In the CHERRY trial, total atheroma burden by IVUS was significantly reduced in patients treated with EPA + statin compared to those receiving statins alone [59]. CCT findings were in line with the prior trial, demonstrating that low-attenuation and fibrofatty plaque decreased after EPA treatment as compared with optimal medical therapy with statins [60]. OCT studies have also showed that EPA added to statin was associated with increased thickness of the fibrous cap as compared with statins in monotherapy [61].

Inclisiran: Inclisiran is a first-in-class small interfering RNA that prevents hepatic synthesis of proprotein convertase subtilisin/kexin type 9, thereby decreasing circulating LDL cholesterol. Its impact on plaque vulnerability and atherosclerotic regression has been evaluated in small series so far [62,63]. Future purposedly designed studies will further assess this issue.

Bempedoic acid: This ATP citrate lyase inhibitor has demonstrated to reduce LDL cholesterol and cardiovascular outcomes in statin-intolerant patients [64]. Thus far, clinical reports have suggested a stabilizing effect in lipid plaques after CCT follow-up [65].

### 4.2. Anti-Inflammatory Drugs

The anti-inflammatory activity of colchicine has been demonstrated to reduce cardiovascular events in patients with chronic CAD [66]. In a cohort study of 80 patients with a recent ACS receiving either colchicine and optimal medical therapy or medical therapy alone and followed by CCT, those patients on colchicine therapy had a greater reduction in low-attenuation plaque volume [67]. A recently published OCT-based clinical trial included patients presenting with ACS and non-culprit lesions exhibiting a lipid-rich pool. Participants were randomized to either colchicine or optimal medical therapy. At 12-months follow-up, those patients receiving colchicine associated a greater regression in plaque-vulnerable characteristics such as TCFA, lipid pool arch, and macrophagic infiltration, along with greater reductions in circulating pro-inflammatory parameters [68].

Lipoprotein-associated phospholipase A2 inhibition has been proposed as a therapeutic target for atherosclerotic vulnerable plaques. A randomized trial reported a reduction in plaque’s necrotic core by IVUS [69], but the main trials powered for clinical endpoints have been negative thus far [70].

### 4.3. Other Agents

Other treatments like antiplatelet drugs are a cornerstone therapy for atherosclerotic disease. Nevertheless, those treatments are not directed towards plaque stability but rather to minimize platelet aggregability. Therefore, despite their efficacy in preventing ischemic events, their current role for treating vulnerable plaques does not extend beyond the global recommendations for patients with coronary artery disease. New drugs exhibiting an impact on cardiovascular outcomes like GLP-1 receptor agonists might also be associated with plaque stability, although further studies are necessary [71].

## 5. Local Invasive Therapies for the Management of Vulnerable Plaques

According to current evidence, percutaneous coronary revascularization is recommended in lesions causing hemodynamically significant flow limitation and in lesions leading to acute coronary syndromes [72]. Based on a recent meta-analysis [73], revascularization reduces the risk of spontaneous myocardial infarction and cardiac death in patients with chronic coronary syndromes, with no significant reduction in all myocardial infarction and all-cause death. Additionally, PCI reduces the incidence of death, cardiac death, and MI in patients with unstable CAD [74].

In patients with STEMI and multivessel disease, the COMPLETE trial demonstrated that elective revascularization of non-culprit lesions with ≥70% diameter stenosis, following successful primary PCI of the culprit lesion, reduced 5-year rates of future ACSs [75]. However, as previously mentioned, events can originate from vulnerable plaques, regardless of whether they are flow-limiting or not [76,77]. Given that vulnerable plaques have been frequently identified in non-culprit lesions in the setting of ACS [78], it could be hypothesized that preventive treatment of these lesions would reduce the residual risk of patients after ACS.

The concept of preventive PCI is based on observational and experimental studies indicating that implantation of metallic stents or bioresorbable vascular scaffold (BVS) leads to a thickening of the fibrous capsule due to neointima formation, shear stress normalization, and lumen enlargement [79,80,81], potentially reducing the risk of rupture of vulnerable plaques. In this scenario, the PROSPECT-ABSORB, a clinical randomized trial, investigated the outcomes of performing preventive PCI on non–flow-limiting vulnerable plaques [82]. Following successful PCI of STEMI culprit lesions, a total of 182 patients with angiographically nonobstructive stenoses exhibiting an IVUS-determined plaque burden of ≥65% were randomized to receive lesion treatment with a BVS combined with guideline-directed medical therapy (GDMT) or GDMT alone. Angiographic follow-up at 25 months showed that the primary endpoint of minimal lumen area (MLA) assessed by IVUS was significantly larger in the BVS group (6.9 mm^2^ vs. 3 mm^2^; *p* < 0.0001). In addition, treatment with BVS reduced the lipid content of the plaque and led to a median neointima hyperplasia of 210 μm compared with GDMT alone. In terms of safety, no significant differences in the target lesion failure rate were observed between the groups at 2 years. Although the trial was not powered for clinical outcomes, randomized lesion-related MACE (a composite of cardiac death, MI, or unstable angina) at 4 years occurred in 4.3% of BVS-treated patients compared with 10.7% with GDMT alone (OR 0.38, 95% CI 0.11–1.30; *p* = 0.12).

The results of the PREVENT trial (Preventive percutaneous coronary intervention versus optimal medical therapy alone for the treatment of vulnerable atherosclerotic coronary plaques) have recently been published [83,84]. This multicenter, open-label, randomized controlled trial aimed to assess the impact of preventive PCI on MACE among patients presenting with high-risk, non–flow-limiting vulnerable plaques identified through intracoronary imaging. After performing PCI on the culprit lesions in patients with ACS and stable CAD, any non-culprit lesions with a stenosis of 50% or more were assessed for functionality using FFR. Non–flow-limiting lesions (FFR > 0.80) underwent further evaluation through intracoronary imaging techniques, including greyscale IVUS, radiofrequency IVUS, a combination of greyscale IVUS and NIRS, or OCT, depending on the judgment of experienced interventional cardiologists. Vulnerable plaques were defined as lesions that satisfied at least two of the following criteria: MLA < 4.0 mm^2^, as measured by IVUS or OCT; plaque burden greater than 70%, as assessed by IVUS; lipid-rich plaque by NIRS ([maxLCBI4mm] > 315); or detection of a TCFA using RF-IVUS or OCT. Eligible patients were randomly assigned to undergo PCI combined with either GDMT or GDMT alone. PCI of the vulnerable plaques was performed using BVS and cobalt-chromium everolimus-eluting metallic stents. The primary outcome was a composite of death from cardiac causes, target-vessel MI ischemia-driven target-vessel revascularization, or hospitalization for unstable or progressive angina at 2 years. A total of 1606 patients were included, without significant differences in baseline characteristics between the two groups. At 2-year follow-up, the primary outcome occurred in three (0.4%) patients in the preventive PCI group and in 27 (3.4%) patients in the GDMT group (HR 0.11, 95% CI 0.03–0.36, *p* = 0.0003), and the effect of preventive PCI was directionally consistent for each component of the primary composite outcome. After a median follow-up of 4.3 years, patients who underwent PCI showed a significantly lower cumulative incidence of the composite primary endpoint, with a 46% reduction in risk observed across the study period (6.5% vs. 9.4%; HR 0.54, 95% CI 0.33–0.87, *p* = 0.009). Additionally, preventive PCI was associated with a reduction in the composite patient-oriented risk of all-cause death, all myocardial infarctions, or any repeat revascularization (HR 0.69, 95% CI 0.50–0.95, *p* = 0.022). Finally, in a post-hoc as-treated analysis, the durability of preventive PCI appeared more sustained with cobalt-chromium everolimus-eluting metallic stents compared to BVS. Figure 2 and Figure 3 summarize the main aspects regarding vulnerable plaque development and progression, the diagnosis pathway for that entity, and the main medical and invasive therapeutic approaches available.

## 6. Future Perspectives

Compelling evidence indicates that vulnerable plaques represent a significant risk for adverse cardiac events, irrespective of whether they cause flow limitation [85,86,87,88] and despite management with optimal medical treatment. The pursuit of identifying vulnerable plaques with the aim of categorizing high-risk lesions and patients prone to future cardiac events, remains a central focus of extensive research. However, given the variability in the progression of individual plaques [89], the value of assessing plaque structure at a single point in time remains uncertain.

There are multiple definitions of vulnerable coronary lesions, with recent emphasis placed predominantly on those established through invasive imaging techniques. However, no intravascular imaging modality is able to identify and evaluate all plaque characteristics associated with vulnerability. Therefore, the combination of intravascular techniques with differential features might be crucial in the near future in order to enhance sensitivity and specificity for vulnerable plaque detection. Currently, only IVUS-NIRS has a relatively significant penetration in clinical practice, and other combinations are expected to gain availability. Besides, the extent to which high-definition IVUS will increase the accuracy for TCFA and lipid-rich plaque characterization and its correlation with histology remain as matters of great interest, which should be elucidated in the future.

Conceptually, these vulnerable plaques could be stabilized with the use of intensive medical therapy. Lipid-lowering drugs have widely demonstrated their local benefits in terms of plaque regression and stabilization, both in the setting of general atherosclerotic disease and more specifically in patients with high-risk plaque features. Noteworthy, OCT studies allowed to demonstrate that plaque composition benefits precede plaque regression in patients receiving high-intensity lipid-lowering therapies (i.e., combination of statins, ezetimibe, and PCSK9i). This provides a rationale that might help understand the early separation in the curves of cumulative incidence for cardiovascular events in PCSK9i trials, favoring those patients receiving more aggressive LDL-lowering therapies. The V-ACCELERATE (NCT06372925) and V-PLAQUE (NCT05360446) studies will use both IVUS and OCT (the former) and CCT (the latter) to evaluate changes in plaque composition and volume with the use of the novel lipid-lowering therapy inclisiran. Besides, more robust data on the actual impact of bempedoic acid in local plaque-related endpoints is expected for the early future. Which combination of drugs among the growing armamentarium of LDL-lowering therapies best fit each patient remains to be addressed. Probably, a combination of clinical and local plaque-oriented features will help in the selection of such therapies, with an upfront strategy of reducing LDL as much and as soon as possible. In regard to other systemic interventions, only colchicine as an anti-inflammatory drug persists as a potential therapy for stabilization of the atherosclerotic process. Ongoing trials based on OCT (ACTRN12618000809235) and CCT (NCT05347316) will shed more light on the benefit of colchicine for local plaque-oriented endpoints.

The field of invasive coronary interventions as preventive therapies for the management of vulnerable plaques is growing exponentially. PCI intervention comes with a trade-off in terms of type 4a myocardial infarction and the risk for stent thrombosis and restenosis but, given the marginal rates of such pitfalls in current practice, this strategy had the rationale to be evaluated for vulnerable plaque management. The publication of the PREVENT trial results may represent a paradigm shift in the management of multivessel disease, suggesting that performing preventive PCI of non–flow-limiting high-risk vulnerable plaques could improve patient outcomes beyond the benefits achievable with medical therapy alone. These results advocate for an expansion of the indications for PCI, including non–flow limiting vulnerable plaques. Furthermore, this trial extends the results to patients with stable CAD, previously under-represented in other trials. This approach, however, implies the need for invasive imaging, suggesting that it is probably only suitable for patients who have already been referred for invasive angiography. In this regard, CCT could be useful as a non-invasive screening tool for stable outpatients who are at increased risk of adverse events due to multiple cardiac risk factors. Further concerns with this preventive PCI strategy arise from the fact that performing and interpretating intravascular imaging for vulnerable plaque detection is not yet a global standard, and therefore operators should receive specific training in order to prevent misclassification of the lesions to be treated.

To strengthen the notion that preventive PCI of vulnerable plaques identified through imaging enhances patient outcomes, several large-scale randomized trials are currently underway. The INTERCLIMA trial (NCT05027984) is evaluating whether a revascularization strategy for non-ACS culprit lesions based on OCT plaque vulnerability criteria outperforms the classical approach using pressure guidewire assessment. Another study, the VULNERABLE trial (NCT05599061), is investigating whether preventive PCI of non-culprit, non–flow-limiting STEMI lesions that meet OCT vulnerability criteria offers superior outcomes compared to GDMT alone. Additionally, the COMBINE INTERVENE trial (NCT05333068) will assess whether PCI based on a combination of FFR and OCT (all FFR ≤ 0.80 and all vulnerable plaques) is superior to PCI with FFR alone. The potential use of drug-coated balloons in this scenario is also under investigation [90]. 

The combination of preventive PCI in addition to advanced pharmacological treatments, including PCSK9 inhibitors and inclisiran to the already widely used statins and ezetimibe, might constitute the cornerstone for coronary events prevention in patients with high-risk atherosclerotic coronary plaques.

Whether to choose a systemic therapeutic approach or a focal PCI strategy remains elusive, but both therapies might be compatible. The PREVENT trial has demonstrated the safety of preventive PCI in vulnerable plaques, but additional data are required to establish its efficacy in reducing future cardiovascular events given the low event rate observed in both study groups during follow-up. Besides, PCI addresses exclusively a specific segment of the vessel, potentially leaving untreated other plaques with vulnerable characteristics that were not assessed by imaging. On the other hand, novel lipid-lowering therapies target the whole vasculature and may benefit all vulnerable plaques despite being previously diagnosed or not. As a downfall, those therapies are currently notably expensive. Implementing a systemic lipid-lowering approach without confirming the presence of vulnerable plaques could result in substantial healthcare expenditures for patients whose actual risk remains uncertain. In our opinion, lipid-lowering drugs should be selected depending on the patient’s overall cardiovascular risk independently of the presence of vulnerable plaques. Focal therapies like PCI need further data, although the implementation of imaging technologies facilitating its diagnosis and extension will facilitate the selection of target vessel segments before the procedure. Cost-effective studies will also help in implementing treatment strategies.

## 7. Conclusions

Advances in imaging techniques have enabled a more accurate detection of vulnerable plaques. Several prospective studies using various intracoronary imaging modalities—such as conventional or radiofrequency IVUS, NIRS-IVUS, and OCT—have demonstrated the ability to identify high-risk plaques with specific imaging characteristics associated with an elevated risk of adverse events over the long term.

Pharmacologic therapies have shown efficacy in promoting atherosclerotic plaque regression. Even though GDMT remains the cornerstone of secondary prevention in patients with established coronary atherosclerosis, the residual risk of adverse events remains unacceptably high. The knowledge gap surrounding the optimal management of vulnerable plaques in non-culprit lesions is gradually being addressed through recently published data suggesting a potential benefit of prophylactic PCI in this context. This approach requires further validation, with additional evidence expected to emerge in the near future.

## Figures and Tables

**Figure 1 jcm-14-01539-f001:**
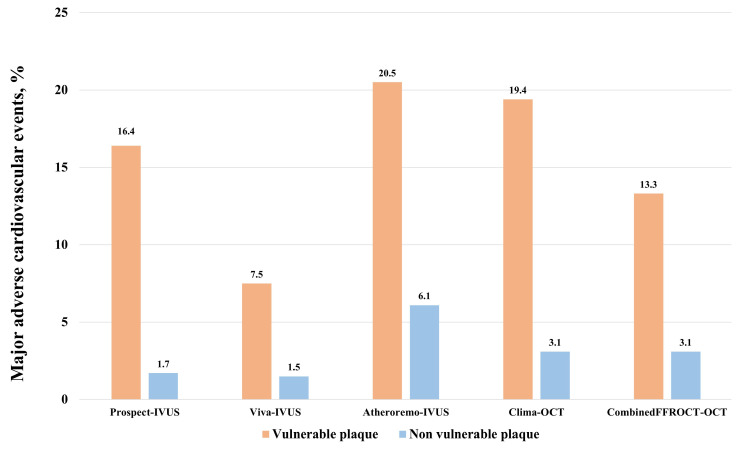
Rates of major adverse cardiovascular events in the main studies evaluating the clinical impact of vulnerable plaques evaluated either with intravascular IVUS or OCT. IVUS: intravascular ultrasound. OCT: optical coherence tomography. Events were evaluated at 3-year follow-up for PROSPECT and VIVA trials, 1-year follow-up for ATHEROREMO and CLIMA trials, and 18-months follow-up for the COMBINEDFFROCT trial.

**Figure 2 jcm-14-01539-f002:**
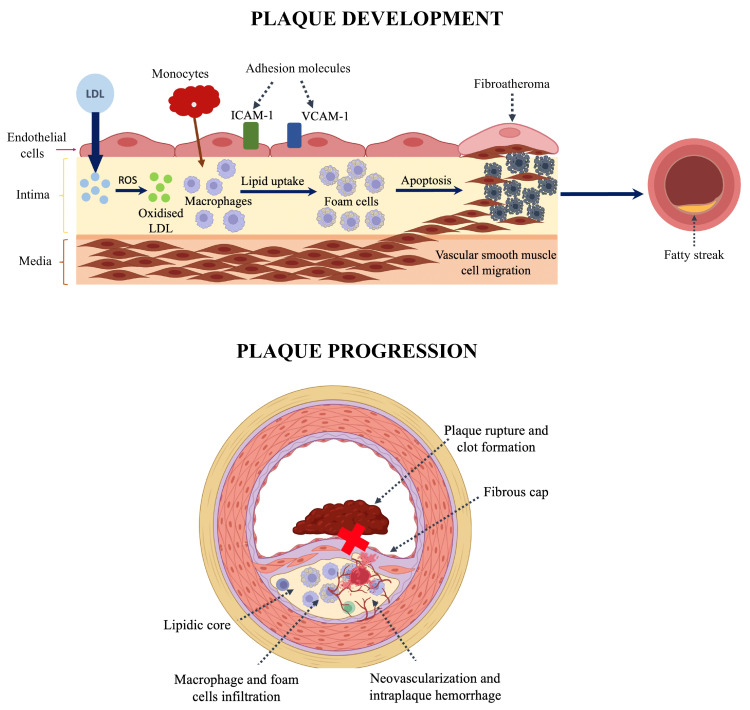
(**Upper panel**): Pathophysiology of atheroma plaque formation: 1. Endothelial dysfunction leads to increased LDL permeability and reduced nitric oxide production. 2. LDL molecules are oxidised in the subendothelial space (ROS: reactive oxygen species), and they promote the upregulation of adhesion molecules ICAM1 (intercellular adhesion molecule 1) and VCAM1 (vascular cell adhesion molecule 1). 3. Monocytes bind to these molecules and migrate into the vessel wall, where they convert into foam cells. 4. Excessive cholesterol uptake can disrupt autophagy regulation, induce cytotoxic effects, and ultimately lead to cell death. Additionally, inflammatory pathways—mediated by pro-inflammatory cytokines such as IL-18 and IL-6, as well as microRNAs—further drive the onset and progression of atherosclerosis. 5. Fatty streak is developed as an early stage of atherosclerosis, with no significant luminal narrowing. (**Lower panel**): Pathophysiology of vulnerable plaque development and progression: 1. Formation of necrotic core due to foam cell apoptosis and necrosis. 2. Thinning of fibrous cap: activated macrophages release matrix metalloproteinases and other enzymes that degrade extracellular matrix. 3. Neovascularization and intraplaque hemorrhage: immature microvessels rupture. 4. Plaque rupture and thrombosis: core exposure triggers thrombus formation.

**Figure 3 jcm-14-01539-f003:**
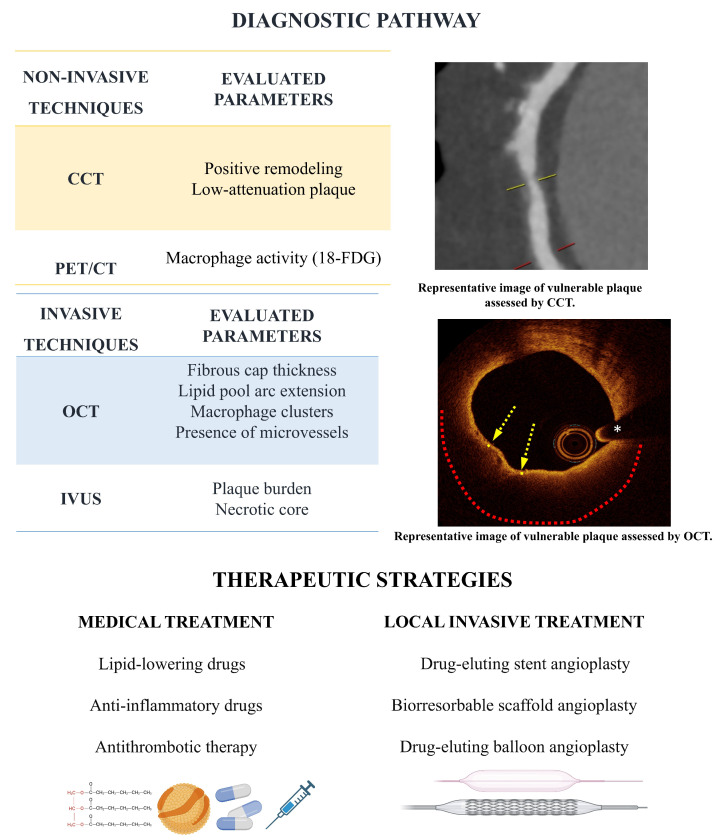
Diagnostic pathway and therapeutic strategies. Low attenuation plaque on CCT indicating vulnerability. On the OCT image the yellow arrows indicate the thin fibrous cap covering a lipid core extending 180° along the circumference (red dotted line). The white asterisk marks the wire artifact.

**Table 1 jcm-14-01539-t001:** Characteristics of the main intravascular imaging techniques, displaying general features and those focused on vulnerable plaque assessment.

	IVUS	OCT	NIRS
**General characteristics**			
**Energy source**	Ultrasound	Infrared light	Near infrared light
**Resolution, πm**	50–150	10	NA
**Frame rate**	30	100	NA
**Imaging through calcium**	No	Yes	Yes
**Vulnerable plaque assessment**			
**Thin fibrous cap**	-	+++	-
**Lipid core**	+	++	++
**Macrophage infiltration**	-	++	-
**Neovascularization**	-	++	-
**Vessel wall remodeling**	++	-	++
**Spotty calcification**	+	+	-
**Plaque rupture**	+	+++	-
**Plaque erosion**	-	++	-
**Thrombus**	+	+++	-

IVUS: intravascular ultrasound. NA: not available. NIRS: near infrared spectroscopy. OCT: optimal coherence tomography. +++: very useful for detection. ++: useful for detection. +: can be detected. -: not useful for detection.

**Table 2 jcm-14-01539-t002:** Main studies evaluating novel lipid-lowering therapies and their impact on coronary atherosclerotic plaque.

Study	Therapy	Sample Size	Clinical Setting	Imaging Method	Therapy Duration	Change in LDL	Change in Atherosclerotic Plaque Features
**Ako et al. [54]**	Alirocumab vs. statin	206	ACS	IVUS	9 months	Relative reduction: -Alirocumab: 64.5% -Control: 7.6%	Total atheroma volume reduction: -Alirocumab 4.8% -Statin 3.1%
**Räber et al. [55]**	Alirocumab vs. placebo	300	ACS	NIRS-IVUS OCT	12 months	Absolut LDL reduction. -Alirocumab: 131.2 mg/dL. -Placebo: 76.5 mg/dL	LCBI_4mm_: -Alirocumab: −79.42 -Placebo: −37.6 Minimal fibrous cap thickness: -Alirocumab: 62.7 πm -Placebo: 33.2 πm
**Nicholls et al. (GLAGOV) [56]**	Evolocumab vs. placebo	968	Stable CAD	IVUS	18 months	Absolut LDL reduction: -Evolocumab: 56.3 mg/dL -Placebo: 0.2 mg/dL	Total atheroma volume: -Evolocumab: 5.8 mm^3^ -Placebo: 0.2 mm^3^
**Nicholls et al. (HUYGENS) [57]**	Evolocumab vs. placebo	161	ACS	OCT	12 months	Absolute LDL reduction: -Evolocumab: 114.2 mg/dL -Placebo: 55.3 mg/dL	Minimal fibrous cap thickness: -Evolocumab: 100.6 πm -Placebo: 81.7 πm Maximum lipid arch: -Evolocumab: 171.9° -Placebo: 193.6°
**Hirai et al. [58]**	Evolocumab + statin vs. statin	98	Stable CAD	CCT	6 months	Absolute LDL reduction: -Evolocumab: 51.1 mg/dL -Control: 11.9 mg/dL	Minimum plaque density: -Evolocumab: 84.9 HU -Control: 44 HU Remodeling index: -Evolocumab: 1.19 -Control: 1.35
**Watanabe et al. [59]**	EPA + statin vs. statin	241	Stable CAD and ACS	IVUS	8 months	Relative reduction -EPA: 23.6% -Control: 17.6%	Total atheroma volume reduction: -EPA: 7.8 mm^3^ -Control: 5.9 mm^3^
**Motoyama et al. [60]**	EPA (low and high dose) + statin vs. statin	210	ACS	CCT	12 months	Absolute LDL reduction: -EPA: 45 mg/dL -Control: 43 mg/dL	Absolut different in fibro-fatty plaque volume: -EPA: −2.82 mm^3^ -Control: +1.97 mm^3^
**Nishio et al. [61]**	EPA + statin vs. statin	341	Stable CAD and ACS	OCT	9 months	Absolute LDL reduction: -EPA: 57.9 mg/dL -Control: 47.1 mg/dL	Increase in fibrous cap thickness: -EPA: 54.8 πm -Control: 23.5 πm Decrease in lipid arch extent: -EPA: 34.4° -Control: 12.7°

ACS: acute coronary syndrome. CAD: coronary artery disease. CCT: coronary computed tomography. EPA: eicosapentaenoic acid. IVUS: intravascular ultrasound. LDL: low-density lipoprotein. NIRS: near infrared spectroscopy. OCT: optical coherence tomography.

## Data Availability

No new data was created.

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
