# Peer review of "Perspectives in the Diagnosis, Clinical Impact, and Management of the Vulnerable Plaque"

_jcm, 2025, doi:10.3390/jcm14051539_

Round 1
Reviewer 1 Report
Comments and Suggestions for Authors
The topic of this peer-reviewed manuscript is definitely interesting and of potential value to “Journal of Clinical Medicine” readers.
In this review „Perspectives in the Diagnosis, Clinical Impact, and Management of the Vulnerable Plaque” by Alberto Alperi et al. the authors aimed to revise the current concept of vulnerable coronary plaque and its repercussion, to summarize the main pharmacological approaches for its management, and to provide an updated overview of the available evidence on percutaneous interventional strategies in this clinical setting.
- Fill the following sections at the end of the manuscript: Author Contributions, Funding, Conflicts of Interest etc.
- Some references are very old; I suggest replacing them if possible.
- Check the use of abbreviations throughout the manuscript.
- Visual representations can make the content more accessible and appealing to readers. Consider including 1) a figure presenting invasive and non-invasive coronary imaging modalities, 2) schematic representation of a suspected vulnerable plaque.
- What about magnetic resonance imaging (MRI) as a non-invasive imaging modality?
Minor revision of the English language required.

Author Response
Responses to the comments from the Reviewer#1:
In this review „Perspectives in the Diagnosis, Clinical Impact, and Management of the Vulnerable Plaque” by Alberto Alperi et al. the authors aimed to revise the current concept of vulnerable coronary plaque and its repercussion, to summarize the main pharmacological approaches for its management, and to provide an updated overview of the available evidence on percutaneous interventional strategies in this clinical setting.
- Fill the following sections at the end of the manuscript: Author Contributions, Funding, Conflicts of Interest etc.
This has been done.
- Some references are very old; I suggest replacing them if possible.
We have changed some of the references for more recent articles. However, some early studies are cited since they are the cornerstones of the field.
- Check the use of abbreviations throughout the manuscript.
This has been rechecked.
- Visual representations can make the content more accessible and appealing to readers. Consider including 1) a figure presenting invasive and non-invasive coronary imaging modalities, 2) schematic representation of a suspected vulnerable plaque.
New images have been added to the revised version of the manuscript following this reviewer´s and reviewer´s 3 suggestions.
- What about magnetic resonance imaging (MRI) as a non-invasive imaging modality?
A paragraph has been added in that matter (page 5, lines 195-199), highlighting its limited use in that context.
Reviewer 2 Report
Comments and Suggestions for Authors
The submitted narrative review is overall well written and conceptualized. It presents a large amount of data on the topic and appears relevant. Language is fine. Please find below specific comments:
- A novel and unexplored topic is the development of novel biomarkers able to differentiate between stable and unstable plaques and/or to predict prognosis (e.g. CD93, PMID: 35166040, PMID: 37371490). This is crucial since several mechanisms are involved including inflammation, apoptosis, oxidative stress etc...).
- Figure 2 is unnecessary and should be replaced with a summarizing figure on the pathophysiological mechanisms of unstable plaque development and progression together with diagnostic and therapeutic pathways.
Author Response
Responses to the comments from the Reviewer #2:
The submitted narrative review is overall well written and conceptualized. It presents a large amount of data on the topic and appears relevant. Language is fine. Please find below specific comments:
- A novel and unexplored topic is the development of novel biomarkers able to differentiate between stable and unstable plaques and/or to predict prognosis (e.g. CD93, PMID: 35166040, PMID: 37371490). This is crucial since several mechanisms are involved including inflammation, apoptosis, oxidative stress etc...).
A novel paragraph has been added to the revised version of the manuscript according to the reviewer´s suggestion (page 5, lines 195-199).
- Figure 2 is unnecessary and should be replaced with a summarizing figure on the pathophysiological mechanisms of unstable plaque development and progression together with diagnostic and therapeutic pathways.
New images have been added to the revised version of the manuscript following this reviewer´s suggestion, and the previous Figure 2 has been deleted.
Reviewer 3 Report
Comments and Suggestions for Authors
This is a nice and succinct overview on vulnerable plaque pathophysiology, diagnosis, and management. The paper is generally well-written with no major flaws.
I only have some smaller remarks that should be discussed, as outlined below:
- I would advise authors to write more on the intricacies and nuances of rupture of vulnerable plaque. Some questions should be addressed and clarified for the reader: for how many deaths is vulnerable plaque rupture deemed as a culprit among patients with cardiac death, at least from the perspective of autopsy studies? What kind of plaques do rupture in terms of angiographic severity and other characteristics?
- I hope that authors can put their own perspective into this narrative review, e.g. what do they think the future will be - will we have potent pharmacotherapies that will be able to stabilize the plaque so to speak permanently or will "prophylactic" stenting be the road? Which of the two approaches are more likely to be cost-effective? - I would shorten the introductory part in which authors describe basics terminology of plaque natural history and pathophysiology.
- What is the role of antithrombotics, other agents such as GLP-1 agonists etc. in the regression or stabilization of fibroatheroma? This should also be briefly discussed.
- With respect to PREVENT Trial, I think that its accompanying editorial in Lancet should be cited as well - Borovac JA. Percutaneous coronary intervention for non-obstructive vulnerable plaques. Lancet. 2024 May 4;403(10438):1724-1725. doi: 10.1016/S0140-6736(24)00488-4. Epub 2024 Apr 8. PMID: 38604207.
- NIRF-IVUS, IVPA-IVUS, and TRFS-IVUS are also invasive methodologies to detect and characterize vulnerable plaques so they should be mentioned.
Author Response
Responses to the comments from the Reviewer #3:
This is a nice and succinct overview on vulnerable plaque pathophysiology, diagnosis, and management. The paper is generally well-written with no major flaws. I only have some smaller remarks that should be discussed, as outlined below:
-I would advise authors to write more on the intricacies and nuances of rupture of vulnerable plaque. Some questions should be addressed and clarified for the reader: for how many deaths is vulnerable plaque rupture deemed as a culprit among patients with cardiac death, at least from the perspective of autopsy studies? What kind of plaques do rupture in terms of angiographic severity and other characteristics?
Thanks for the comment. This information has been incorporated in the revised version of the manuscript (page 2, lines 73-76).
- I hope that authors can put their own perspective into this narrative review, e.g. what do they think the future will be - will we have potent pharmacotherapies that will be able to stabilize the plaque so to speak permanently or will "prophylactic" stenting be the road? Which of the two approaches are more likely to be cost-effective?
A new paragraph has been added in that sense (page 12 lines 493-509).
-I would shorten the introductory part in which authors describe basics terminology of plaque natural history and pathophysiology.
This has been done accordingly.
-What is the role of antithrombotics, other agents such as GLP-1 agonists etc. in the regression or stabilization of fibroatheroma? This should also be briefly discussed.
This has been discussed in the revised version of the manuscript (page 8, lines 310-318).
-With respect to PREVENT Trial, I think that its accompanying editorial in Lancet should be cited as well - Borovac JA. Percutaneous coronary intervention for non-obstructive vulnerable plaques. Lancet. 2024 May 4;403(10438):1724-1725. doi: 10.1016/S0140-6736(24)00488-4. Epub 2024 Apr 8. PMID: 38604207.
The editorial has been cited according to the reviewer´s suggestion.
-NIRF-IVUS, IVPA-IVUS, and TRFS-IVUS are also invasive methodologies to detect and characterize vulnerable plaques so they should be mentioned.
This comment has been added to the revised version of the manuscript (page 4, lines 133-136).